# Rare Case of Intravascular Myopericytoma—Imaging Characteristics and Review of the Literature

**DOI:** 10.3390/diagnostics12102473

**Published:** 2022-10-13

**Authors:** Simona Manole, Roxana Pintican, Viorel Manole, Cosmin Rusneac, Calin Schiau, Ioana Bene, Carolina Solomon, Sorin Dudea

**Affiliations:** 1Department of Radiology, “Niculae Stancioiu” Heart Institute, Motilor Street, n. 19-21, 400001 Cluj-Napoca, Romania; 2Department of Radiology, University of Medicine and Pharmacy “Iuliu Hatieganu” Cluj-Napoca, Babes Street, nr. 8, 400000 Cluj-Napoca, Romania; 3Department of Cardiovascular Surgery, “Niculae Stancioiu” Heart Institute, Motilor Street, n. 19-21, 400001 Cluj-Napoca, Romania; 4Department of Radiology, Emergency Clinical County Hospital Cluj-Napoca, Clinicilor Street, n. 3-5, 400006 Cluj-Napoca, Romania

**Keywords:** myopericytoma, intravascular, arterial wall, brachial artery

## Abstract

Myopericytoma is a rare vessel wall tumor, a subtype of hemangiopericytoma that usually develops subcutaneously. Intravascular myopericytoma is a rarer subtype, with only few cases reported in the literature and even fewer with imaging modalities included. We report the case of a 36-year-old man who was referred to our institution with a painless, palpable mass in the right arm and was evaluated with MRI, grey-scale and Doppler-mode ultrasound. Tumor histopathology and imaging characteristics are presented together with the role that each imaging modality played in the management of the patient.

## 1. Introduction

Myopericytoma is an uncommon, benign tumor, the most common subtype of hemangiopericytomas [1]. Due to the fact that various terms were previously used when reporting them, their real incidence is still unknown. The majority of myopericytoma tumors are slow-growing tumors with benign behavior, but up to 20% of them may pursue a malignant outcome with recurrent disease and distant metastasis [2]. Typically located in the dermis and superficial soft tissue, myopericitoma is almost never encountered within a blood vessel wall. 

Intravascular myopericytoma (IVMP) is a rare subtype of myopericytoma, that was recognized as a distinct entity by the World Health Organization only in 2002 [3]. There were conflicting and overlapping pathology features reported over the years with myofibroma and more important with sarcoma, but generally the tumor was contained within a blood vessel The imaging characteristics of myopericytoma and IVMP are rarely reported in literature, and only one review is available on breast myopericitoma [4]. 

Therefore, we aim to present an extremely rare location of intravascular myopericitoma, contained within the arterial wall (IVMP) and not the blood vessel lumen as the majority of previously published cases; furthermore, we provide a review of the imaging characteristics all reported IVMP, imaging characteristics of myopericytoma and, last, we discuss and suggest an algorithm for the imaging evaluation of IVMP patients.

## 2. Case Report

We report the case of a 36-year-old man who was referred to our institution with a painless, palpable mass in the right arm, close to the right cubital fossa, which had been slowly growing for the past 5 years. The mass was mobile, soft, pulsating and non-tender to touch. No overlying skin or soft tissue changes were noted and the ulnar and radial artery pulses were present.

MRI highlighted the vascular origin of the mass, arising from the arterial wall, located above the brachial artery bifurcation, anteriorly to the right medial epicondyle. It was well delineated with circumscribed borders and had T2 hyperintense signal intensity (SI), T1 hypointense SI (Figure 1A) and avid, homogenous enhancement after contrast administration (Figure 1B). Small vessels were seen originating from the brachial artery with intratumoral distribution.

US showed a solid mass surrounding the brachial artery, located in the arterial wall, with a clear delimitation between the internal and external wall of the artery (Figure 2). The mass was encasing the artery’s lumen without stenosis at Doppler US (Figure 3A). The brachial vein and nerve were situated nearby without evident signs of tumor invasion nor growth around them. The solid mass had internal vascularity, with a small nutritive vessel originating from the brachial artery (Figure 3B). 

Surgical resection (Figure 4) with a transverse incision at the level of the right cubital fossa was the management of choice. Intraoperatively, a fusiform pulsating tumoral mass of 6.5/3.5 cm was seen, surrounding the brachial artery. The artery was clamped proximal and distal to the tumor, with total tumor resection with an inversed right internal saphenous vein graft used to restore the arm’s blood supply. No intra- or post-operative complications were noted.

The pathology report showed a macroscopic well-defined mass with microscopic features of a vascular nodular proliferation. The mass developed within the arterial wall and had a small extension into the adipose tissue. The tumor was composed of medium-caliber vessels surrounded by myoid cells, well-represented smooth muscular component and hyaline nodules with elongated cells. Immunohistochemistry highlighted a very weak mitosis index of 5–6 per 50 HPF, Ki 67% proliferative index of 5%, positive alpha actin and desmin and negative S100; cytokeratin AE1/3 was inconclusive and CD31 and CD34 were positive at the endothelial level (Figure 5). The final diagnosis was of myopericytoma with cavernous angio-leiomyomatous component and focal atypical myoid stroma. 

The patient had not undergone any adjuvant treatment and the follow-up US imaging at 5 years was clear of tumor relapse. 

## 3. Discussion

We presented a rare case of myopericytoma located within the brachial artery wall, discuss histopathology characteristics and highlight the role of different imaging modalities (US, Doppler US and MRI) in the diagnostic and characterization of this peculiar tumors.

Myopericytoma is a vascular tumor originating from myopericytes, with benign behavior in most of the cases. It originates in the cells named “myopericytes” that were relatively recently described (in 1992) as transitional cells between pericytes and smooth muscle cells of blood vessels. They are specialized contractile cells surrounding the endothelial cell layer of small vessels, reason why myopericytoma develops encasing the artery’s lumen.

The term “myopericytoma” or “perivascular myoma” was used for the first time in 1998, by Granter et al. to describe classical myopericytoma. In 2002, the World Health Organization (WHO) recognized this tumor as a distinct entity [5].

It may occur in various age groups, reported as ranging from 10 to 87 years; however, most are seen in adults with a median age of 49 years [4] with a slight male predilection [5]. The most common location is in the dermis or subcutaneous tissue of the extremities in adults, but very rare visceral or deep soft tissue locations have been reported [5,6,7]. The lower extremities are most frequently affected followed by the upper extremity, the head and neck area and the trunk. The majority of lesions are solitary but multicentric presentation involving the same or different anatomic locations may also be observed [8,9].

Intravascular myopericitoma (IVMP) is an extremely rare subtype of myopericitoma, where the tumor is contained within a blood vessel wall, with or without invasion into surrounding structures.

We performed PubMed database research using MESH terms as “blood vessel tumor”, “myopericitoma” and also non-MESH terms as “intravascular myopericitoma”, “IVMP” as key words, with 15 papers reported in the English literature, and 10 actually including IVMP characteristics [10,11,12,13,14,15,16,17,18,19]. The remaining 5 were broad reviews including already reported cases, thus, we excluded them.

The first case of IVMP was reported in 2002, by McMenamin et al. as a “long-standing myopericytoma that was entirely located within the lumen of a subcutaneous vein” [10]. Few case reports followed McMenamin et al., and fewer had imaging characteristics included.

Overall, the authors reported solid lump masses of the arms and legs, mouth and orbital area. Patients were symptomatic and followed surgery with histopathology diagnosis. Compared to general myopericitoma which may or may not be painful, all patients with IVMP complained of long-lasting pain. McMenamin et al., suggested the associated intravascular thrombus to be the cause for the pain in these patients [10].

Furthermore, two authors reported US and two MRI features of IVMP. No author reported computed tomography characteristics of the IVMP.

Regarding the IVMP characteristics, at US, all masses were well-defined, solid, with internal Color Doppler signal corresponding to internal vascularity. Tumor vascularity was also depicted on MRI, as homogeneous enhancement after contrast administration.

An accurate location of the IVMP was reported in cases investigated by US, which was able to highlight the tumor mass within the blood vessel wall. In cases with MRI examination, Mahapatra et al. and Xia et al. concluded that MRI was unable to identify the intravascular origin of the tumor in both cases. In our case, the MRI provided relevant information regarding the exact location of the tumor, within the arterial wall, and helped excluding a potential malignant lesion (homogeneous mass with no invasion into surrounding structures). First possible explanation could be the improvements on MRI sequences (higher resolution and contrast) as the previously reported cases were almost 20 years old; second, the tumor size, as one MRI case failed to located the millimetric tumor within a digital artery of the finger. Thereby, we support the role of MRI could have in characterizing this peculiar tumor masses Table 1.

To the best of our knowledge, there is no available review of the imaging modalities used in the diagnostic diagnose and characterization of myopericytoma, a brief review being available only on breast myopericytoma due to its possible misdiagnosis with breast cancer [4]. Authors concluded that myopericitoma can be differentiated from breast cancer and other hypervascular lesions originating within the breast, by US and breast MRI. The main differentials for myopericytoma of the breast, should include Phyllodes tumors, cellular fibroadenomas and breast papillary carcinomas. The breast myopericytoma showed circumscribed margins, hypointense on T1 and hyperintense on T2-weighted sequences. An avid enhancement and a type II curve (Kuhl curve) were reported. No diffusion weighted imaging coefficient or apparent diffusion coefficient (DWI/ADC) characteristics were mentioned. Rarely, breast cancer may display a hyperintense T2-weighted appearance and a type II enhancement curve, but it usually shows restricted diffusion (hyperintense on DWI and hypointense on ADC). However, this is encountered in rare breast cancer subtypes, as pure mucinous carcinoma or no-special-type carcinoma with mucinous component [20].

The majority of imaging characteristics of the myopericytoma reported in the literature, are represented by case reports. The most frequently affected areas include superficial soft tissue of the distal extremities, followed by head and neck, urinary tract and female genital tract. In most of the cases, authors have focused on clinical, histology and immunohistochemistry findings, and less was reported about imaging characteristics or the role of imaging in their diagnostic.

Thus, we expanded our search to include all relevant papers which reported imaging modalities of myopericitoma [21,22,23,24,25,26,27,28,29]. We found that some of the features are common between myopericytoma and its rarer subtype, IVMP.

On US, all the masses had well-defined margins, hypoechoic echogenicity and homogeneous ecostructure, with internal vascularity exhibited on Doppler modalities.

CT showed homogeneous masses on both native and postcontrast sequences, and was helpful in delineating the tumor’s margin and invasion into surrounding structures. Some authors reported myopericytoma as multiple solid masses, with circumscribed margins, within the breast and parotid gland and parapharyngeal spaces [20,22].

The MRI angiography protocol varies between different vendors, but generally includes post-contrast sequences acquired with thin slices (1–2 mm), which allows multiplanar reformations (MPR) in different planes [30,31]. Near isotropic voxels with minimal interslice spacing help to generate reformats with low artefactual distortion. TE and TR are minimized to produce T1WI and to shorten scan time. Dynamic 3D angiographic sequences in coronal place, TRICKS (Time Resolved Imaging of Contrast Kinetics) images were obtained with arterial, venous and late multiphase acquisition. Fat suppression further enhanced signal-to-background contrast.

As regards to tumor features, MRI was in concordance with US and CT, showing soft-tissue masses, well-defined on both T1 and T2-weighted sequences. The majority of the masses appeared hypointense on T1-weighted and hyperintense on T2-weighted sequences. Enhancement was reported without any further clarifications in terms of the time-intensity curves or homogeneity.

The FDG-PET/CT may be used in cases of malignant myopericytoma, for the detection of distant metastatic disease. One author reported a myopericytic lesion of the ear, that was moderately positive for GLUT-1 and identified as being hypermetabolic with intensely increased FDG uptake.

Furthermore, we suggest a stratified diagnostic algorithm, based on the available literature, with US as the first line imaging modality, followed by CT and MRI as second-line imaging tools, and least, FDG-PET/CT for selected cases with proved malignancy. (Table 2).

For our patient, US was the second-line imaging modality used to depict and characterize the mass. It helped not only to highlight the mass origin, but also to assess the vessel permeability. The nutritive arterial tumor vessel was well-depicted by the Doppler mode, which suggested an underling tumor process over an arteritis and had an important role in planning the surgery procedure.

Further, we opted for MRI because it provides better soft tissue contrast than CT, accurate relations with surrounding structures and may also indicate towards a low- or high-grade tumor. Myopericytoma is a well-defined, circumscribed mass, non-invading into surrounding structures, with avid and homogeneous contrast enhancement. Generally, a spiculated, infiltrating mass with heterogeneous with non-enhanced necrosis areas, are more suggestive of high-grade tumors.

Regarding the histopathology, literature showed the possibility that these lesions may be malignant, and highlighted the role immunohistochemistry has in diagnosis these tumors. Some authors suggest that myopericytoma is a unifying term for a spectrum of tumors that show overlapping features with myofibromas. The main differentials should include myofibromatosis, archetypal myofibroma, glomangiopericytoma and myofibroma. However, the most important differentiation to make is against sarcoma, malignant and very aggressive tumor. The authors highlight the importance of the growth pattern in making the final histopathology diagnosis.

Furthermore, myopericitoma is positive for muscle-specific actin, SMA, h-caldesmon, and focally for desmin, but is negative for S-100 protein, CD31 antibody, CD34 antibody, factor VIII, human melanoma black 45, and cytokeratin AE1–AE3.

Unlike myopericytoma, authors reported IVMP to be negative for desmin [12]. However, our case stained positively for desmin, and is consisted with only one reported case by McMenamin et al., supporting the possibility of IVMP to present in both forms: positive or negative for desmin.

Biopsy is not indicated in well-defined, benign IVMP, as bleeding can be a common complication. For selected cases with ill-defined borders, where malignancy is suspected, biopsy should be discussed with a multidisciplinary team so that, if performed, the following surgery will include the biopsy trajectory.

The treatment of IVMP is surgical excision with vessel reconstruction when needed.

Recurrence is a rare but possible evolution, with only 2 cases reported in Mentzel et al. series out of 23 lesions with marginal or incomplete excisions. However, recurrence is more frequently encountered than distant metastasis, reported in 5 cases by McMenamim et al. and 2 out of 54 by Mentzel et al. [10,23].

Follow-up with US as the first-line imaging tool, may be a good option, relying on MRI if distant metastases are suspected.

In conclusion, IVMP are rare subtypes of myopericitoma, with few cases reported in the literature. Immunohistochemistry may be positive for desmin, contrary to previously published papers. US should be used as the first-line imaging modality to detect the mass and asses the blood vessel. MRI is a second-line imaging tool which helps delineate the tumor boundaries related to surrounding structures, as well as planning the surgery. Follow-up US imaging must be considered due to the fact that tumor may relapse.

## Figures and Tables

**Figure 1 diagnostics-12-02473-f001:**
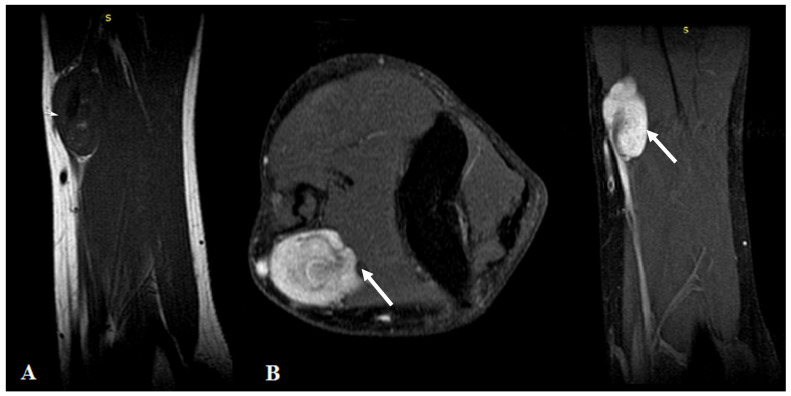
(**A**): MRI imaging—T1WI Sagittal plane showed a homogeneous dark T1 signal of the mass (arrows). (**B**): Contrast-enhanced MRI imaging with fat saturation—axial and sagittal plane showed intense homogenous contrast enhancement of the tumor.

**Figure 2 diagnostics-12-02473-f002:**
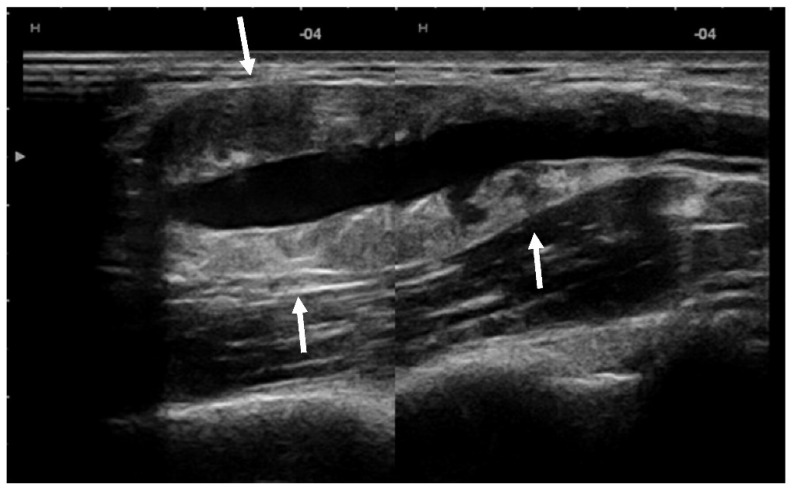
**US** showed a solid mass (arrows) located within the brachial artery wall, encasing the vessel.

**Figure 3 diagnostics-12-02473-f003:**
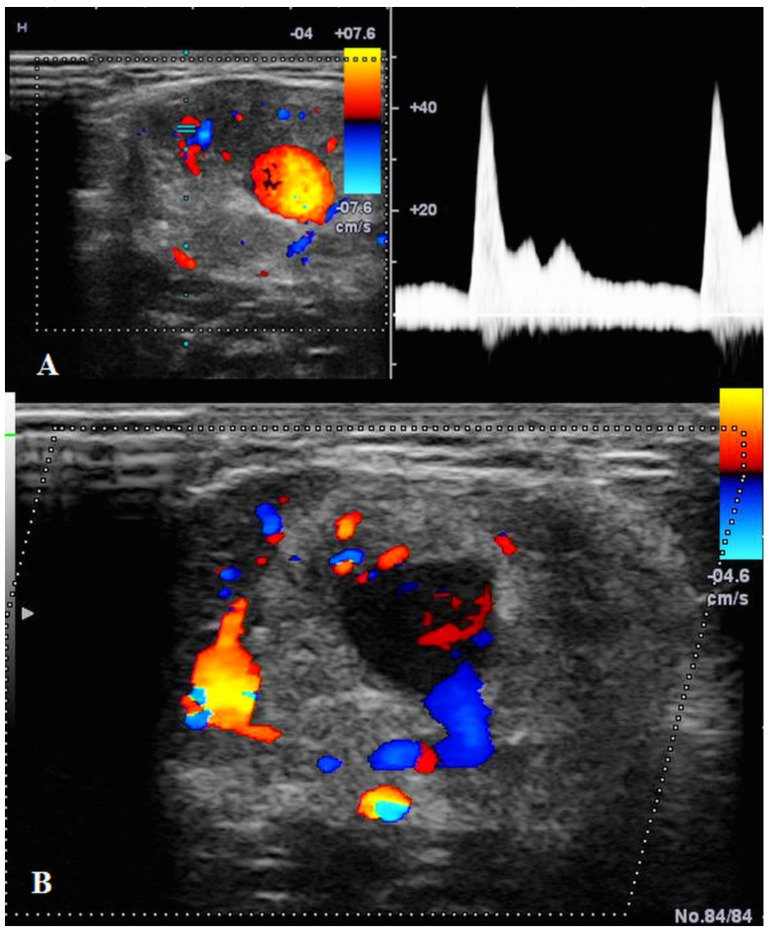
(**A**): **Doppler US** showed no stenosis of the brachial artery with good blood flow. (**B**): **Doppler US** showed the mass having a small nutritive artery originating in the brachial artery.

**Figure 4 diagnostics-12-02473-f004:**
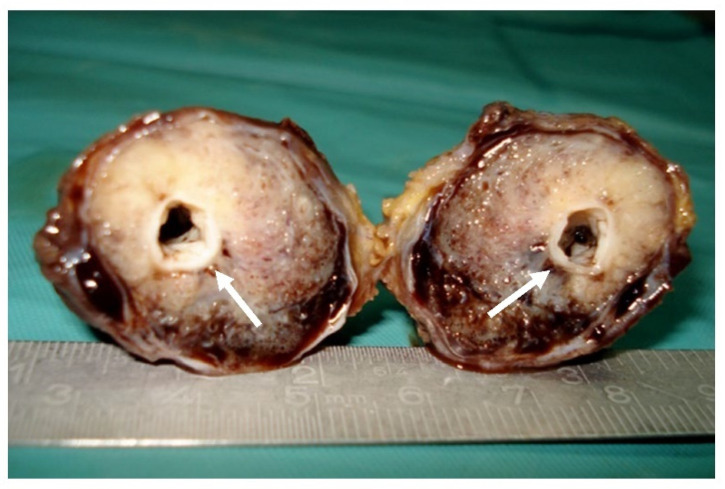
**Surgical specimen:** Gross macroscopic appearance of the tumor with central patent arterial lumen (arrows).

**Figure 5 diagnostics-12-02473-f005:**
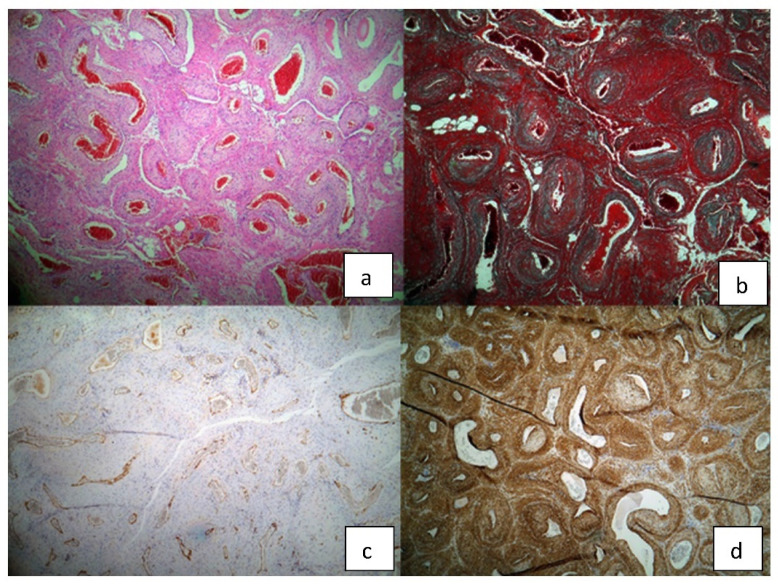
**Histopathological findings of the surgical specimen**: (**a**) H/E staining (**b**) trichrome Masson staining (**c**) CD 31 (**d**) alpha actin.

**Table 1 diagnostics-12-02473-t001:** Intravascular myopericitoma—site and imaging modalities used in diagnostics.

Author, Year *	Age/Sex	Blood Vessel	Site	Imaging Modality
McMenamin et al., 2002	Man/54	Artery	Thigh	NA
Woolard Ac et al., 2007	Man/54	Artery	Hand	NA
Ide F et al., 2007	Woman/45	Artery	Oral	NA
Xia CY et al., 2009	Man/50	Vein	Thigh	MRI
Park HJ et al., 2010	Woman/9	Vein	Infraorbital	NA
Ko Jy et al., 2011	Man/67	Vein	Thigh	NA
Mahapatra P et al., 2014	Woman/59	Artery	Hand	MRI
Valero J et al., 2015	Man/48	Vein	Foot	US
Augusti j et al., 2016	Woman/63	Arterio-venous malformation	Foot	NA
Kagoyama K et al., 2020	Man/78	Vein	Foot	US

* References [10,11,12,13,14,15,16,17,18,19]. NA = not mentioned; MRI = magnetic resonance imaging; US = ultrasound.

**Table 2 diagnostics-12-02473-t002:** The role of imaging modalities used to asses myopericitoma and main imaging features reported in the literature.

Authors/Year	Imaging Modality	Role of Imaging Modalities and Tumor Characteristics
Koh Fong Seen/2014	CT	N/A Avid, homogeneous enhancement
Peters K et al./2018		N/A
	MRI	- Ill-defined, nodular mass with T2 hyperintense SI and avid enhancement suggestive of myopericitomatosis
		- Blood products
		- Surrounding edema
	CT	- Small foci of calcifications
PeiPei Yang et al./2020 *		Both can provide morphological details
	CEUS	- Irregular non-enhancement mass
	MRI	- Avid, heterogeneous enhancement
Manole et al./2022	US	First-line imaging tool
		- detection and characterization
		- assessment of the vessel
		- follow-up
	MRI	Second-line imaging tool
		- boundaries and relation with surrounding structures
		- +/−invading features
		- soft tissue edema
		- distant metastasis

* Myopericytoma of the breast; N/A = not mentioned; CEUS = contrast enhanced ultrasound; +/− = presence or absence.

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
