# Peer review of "Rare Case of Intravascular Myopericytoma—Imaging Characteristics and Review of the Literature"

_diagnostics, 2022, doi:10.3390/diagnostics12102473_

Round 1

Reviewer 1 Report

Dear Authors,

thank you very much for the submitted case report. You describe a very exciting and rare case in a good linguistic and structural quality. I have made some comments that you should please revise:

Line 19: Remove the space 

Line 38: please describe the margins of the tumor. I think it is pseudocapsulated

Line 40: please delete „SI“. Hypointense SI sounds imprecise

Figure 1: please make the arrow a bit bigger AND state which sequences have been used. The latter should be given also in detail in the manuscript.

Figure 3B: please add an arrow to indicate the nutritive artery

Figure 5: please add the letters of the caption (a,b,c,d) in the figure

Line 90/91: please be uniform with the used tense 

Line 114: Better use MESH Terms. If you did please change the wording.

Line 117: please be uniform with italic letters while referencing papers „xxx et al“. E.g. in line 223 you use italic letters.

Line 127: Any idea why? Please comment on the fact. Are there situations in which CT could be helpful?

Line 137/138: Which sequences do you mean here? Please add some information of appropriate MR imaging protocols.

Line 205: should be „in the diagnostics of…“

line 215: the reference numbers should not be superscripted

line 229/230: repetition of „help“

References: there are inconcistencies regarding the reference style. Please be uniform here according to the journal‘s reference style. E.g. you state  the DOI not at all references

Reference 24: Incomplete. Please add the missing information.

Reviewer 2 Report

The authors describe the rare case of a an intravascular myopericytoma with a subsequent review of the literature. It includes imaging characteristics of the presented patient as well as information from the literature and discusses also the pathological features. After a few improvements it should be considered for publication:  

Line 5: please delete „were“

Line 20: Spelling mistake – „myopericytoma“

Line 4 & line 20: please clarify if it is a vessel wall tumor or a tumor that is rarely found in vessel walls

Line 27: please include the reference you refer to

Figure 2: please add arrows for explanation of the US-findings

Figure 3B: please mark the nutritive artery for better clarity

Line 67: was the decision for surgery based on a board discussion or based on the presented literature?

Line 79-80: mitosis index and Ki67 proliferation index are different analyses, please clarify

Line 84: please discuss the decision against adjuvant treatment

Table 1: do you mean thigh?

Line 173: please include the reference for CT characteristics

Line 182: Are these tumors FDG-avid? Or only the malignant ones?

Consider including a figure with a decision tree for the algorithm

Are there recommendations for follow up intervals?

Please add a short discussion on biopsy before surgery

Round 2

Reviewer 1 Report

Dear Authors,

thank you for your revised manuscript. You have improved the casereport but there are some more aspects you should revise:

I asked you to revise the references. The re are still inconsitencies, e.g. you state the PMID sometimes, sometimes not. Furthermore the reference style after reference 22 is varying from the ones before.

Please give more details regrding appropriate MR protocols. Especially dynamic/time resolved MR angiography should be advantageous. Please refer to already published literature here.

Figure 5: Please change the format of the letters according to the other figures.

Table 2, line 5: It seems to be "Thigh" and not "Tight"

Best regards
Your reviewer

Author Response

Dear Reviewer,

Thank you very much for your time and efforts. We addressed carefully your comments and hope we reached your expectations:

  1. We provided the PMID for articles without DOI (as you kindly asked to provide DOI for the first-round review). Now all the references are without PMID. Furthermore, the used the same style for references 22- 28.
  2. The MR protocol and 2 more references were included in the discussion section, as follows: "The MRI angiography protocol varies between different vendors, but generally includes postcontrast sequences acquired with thin slices (1-2mm), which allows multiplanar reformations (MPR) in different planes [30, 31]. Near isotropic voxels with minimal interslice spacing help to
    generate reformats with low artefactual distortion. TE and TR are minimized to produce T1WI
    and to shorten scan time. Dynamic 3D angiographic sequences in coronal place, TRICKS (Time
    Resolved Imaging of Contrast Kinetics) images were obtained with arterial, venous and late multiphase acquisition. Fat suppression further enhanced signal-to-background contrast. ".
  3. All the manuscript text, including the figure captions are written using Times New Roman (including figure 5). Please forgive us, but we are not sure what we should modify. 
  4. On line 5, the word "tight" is corrected.

Best regards,

The authors